# DPP-TTS: Diversifying prosodic features of speech via determinantal point processes

**Seongho Joo**
Seoul National University
seonghojoo@snu.ac.kr

**Hyukhun Koh**
Seoul National University
hyukhunkoh-ai@snu.ac.kr

**Kyomin Jung**[†]
Seoul National University
kjung@snu.ac.kr

## Abstract

With the rapid advancement in deep generative models, recent neural Text-To-Speech (TTS) models have succeeded in synthesizing human-like speech. There have been some efforts to generate speech with various prosody beyond monotonous prosody patterns. However, previous works have several limitations. First, typical TTS models depend on the scaled sampling temperature for boosting the diversity of prosody. Speech samples generated at high sampling temperatures often lack perceptual prosodic diversity, thereby hampering the naturalness of the speech. Second, the diversity among samples is neglected since the sampling procedure often focuses on a single speech sample rather than multiple ones. In this paper, we propose DPP-TTS: a text-to-speech model based on Determinantal Point Processes (DPPs) with a new objective function and prosody diversifying module. Our TTS model is capable of generating speech samples that simultaneously consider perceptual diversity in each sample and among multiple samples. We demonstrate that DPP-TTS generates speech samples with more diversified prosody than baselines in the side-by-side comparison test considering the naturalness of speech at the same time.

## 1  Introduction

In the past few years, Text-To-Speech (TTS) models have made a lot of progress in synthesizing human-like speech (Li et al., 2019; Ping et al., 2018; Ren et al., 2019; Shen et al., 2018; Kim et al., 2021). Furthermore, in the latest studies, several TTS models made high-quality speech samples even in the end-to-end setting without a two-stage synthesis process (Donahue et al., 2021; Kim et al., 2021). Based on these technical developments, TTS models are now able to generate high-fidelity speech.

Meanwhile, human speech contains diverse prosody patterns regarding intonation, stress, and

---
[†] Corresponding author

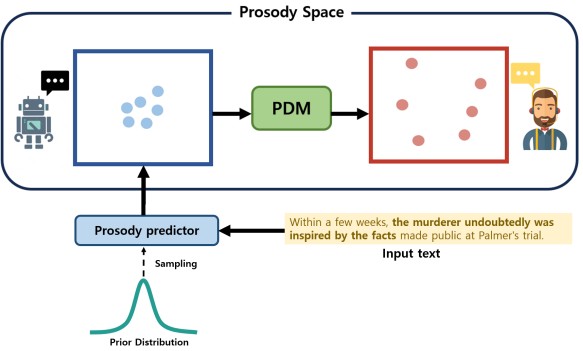

Figure 1: Overview of our DPP-TTS model. Along with the input text segmentation, our Prosody Diversifying Module(PDM) generates a more diverse and smooth prosody pattern compared to the baseline.

rhythm beyond the fidelity of speech. To reflect such acoustic features on generated speech, there have been many attempts to synthesize speech with rich and diverse prosodic patterns. One of the widely used approaches for prosody modeling is to exploit generative models like VAEs and flow models (Hsu et al., 2019; Lee et al., 2021; Ren et al., 2021b; Valle et al., 2021; Vallés-Pérez et al., 2021). These generative TTS models control the extent of variation in speech by sampling prior distribution with adequate temperatures. In other works, auxiliary features such as syntax information and text semantics from BERT embeddings are used to enhance the prosody of speech (Ye et al., 2022; Xu et al., 2020).

However, previous approaches are subject to several limitations. First, sampling with high temperatures for generating diverse prosody patterns often severely degrades the naturalness of speech. In the experiment section, we show that previous TTS models often fail to generate diverse and smooth speech samples to the listeners in diverse sampling temperature settings. Second, previous works for prosody modeling treat each sample independently, thereby not guaranteeing the diversity among separately generated samples.

In this paper, for generating diverse speech samples, we resolve the two aforementioned limitations by adopting Determinantal point processes (DPPs), which have typically shown great results in modeling diversity among multiple samples in various machine learning tasks such as text summarization (Cho et al., 2019) and recommendation systems (Gartrell et al., 2021). We devise a novel TTS model equipped with the new objective function and prosody diversifying module based on DPPs. Through our adaptive MIC objective function and DPP kernel, we can effectively generate speech samples with diverse and natural prosody. However, with the standard DPP, it is challenging to model a fine-grained level of prosody for non-monotonous speech. For more sophisticated prosody modeling, we also propose conditional DPPs which utilize the conditional information for the sampling. Specifically, we first segment the whole input text into more fine-grained text segments, and then sample prosodic features for the targeted segment, reflecting its neighbor segments to generate more expressive samples.

In the process of adopting conditional DPPs into our TTS models, two technical difficulties must be handled. First, the range of targets for the sampling of conditional DPP is ambiguous. Second, prosodic features usually vary in length. We resolve the first issue by extracting key segments from the input text using Prosodic Boundary Detector (PBD). Next, we resolve the second issue by adopting the similarity metric, soft dynamic time warping discrepancy (Soft-DTW) (Cuturi and Blondel, 2017), among the segments of variable lengths.

Experimental results demonstrate that our training methodology becomes stable with the new adaptive MIC objective function, overcoming the instability of sampling-based learning. In addition, our DPP-TTS is highly effective at generating more expressive speech and guaranteeing the quality of speech. We find that our model consistently gains a higher diversity score across all temperature settings than baselines.

In summary, our contributions of the paper are as follows:

- We show that the fine-grained level of the prosody modeling method based on PBD contributes to more diverse prosody in each generated speech sample.

- We generate more diverse prosody patterns

from a single utterance by adopting conditional DPPs into our TTS model with the novel objective function.

- We evaluate and demonstrate that our model outperforms the baselines in terms of prosody diversity while showing more stable naturalness.

## 2 Background

### 2.1 Determinantal point processes

Given ground set $\mathcal{Y}$ which consists of prosodic features in our case, a determinantal point processes (DPP) defines a probability distribution for all subsets of $\mathcal{Y}$ via a positive semi-definite matrix $\boldsymbol{L}$ as followings:

$$\mathcal{P}_{\boldsymbol{L}}(Y) = \frac{\det(\boldsymbol{L}_Y)}{\det(\boldsymbol{L} + \boldsymbol{I})}, \qquad (1)$$

where $\det(\cdot)$ is the determinant of a matrix, $\boldsymbol{L}_Y$ denotes the submatrix of $\boldsymbol{L}$ whose entries are indexed by the subset $Y$ and $\det(\boldsymbol{L} + \boldsymbol{I})$ in the denominator acts as a normalization constant. To model diversity between items, the DPP kernel $\boldsymbol{L}$ is usually constructed as a symmetric similarity matrix $\boldsymbol{S}$, where $S_{ij}$ represents the similarity between two items $x_i$ and $x_j$. Kulesza and Taskar (2010) propose decomposing the kernel $L$ as a Gram matrix incorporating a quality vector to weigh each item according to its quality by defining the kernel matrix as $L_{i,j} = q_i \cdot S_{ij} \cdot q_j$ where $q_i$ denotes the quality of the item. The quality can be chosen as the likelihood of the prosodic feature in our context. Given two prosodic feature $Y_1 = \{i, j\}$, the probability $\mathcal{P}_L(\{i, j\})$ to sample two prosodic features is proportional to $\det(\boldsymbol{L}_{Y_1}) = q_i^2 \cdot q_j^2 \cdot (1 - S_{ij}^2)$. Therefore, two prosodic features are unlikely sampled together if they are highly similar to each other. In contrast, they are more likely sampled together if they have high quality values.

### 2.2 Conditional determinantal point processes

If DPPs are used for diversifying prosodic features corresponding to the target utterance, it would result in diversity among generated samples. However, there still can be monotonous patterns in each generated utterance. To resolve this issue, it is necessary to model the prosodic features of the target accounting into the neighboring segments of the target. In previous studies, DPPs are extended to

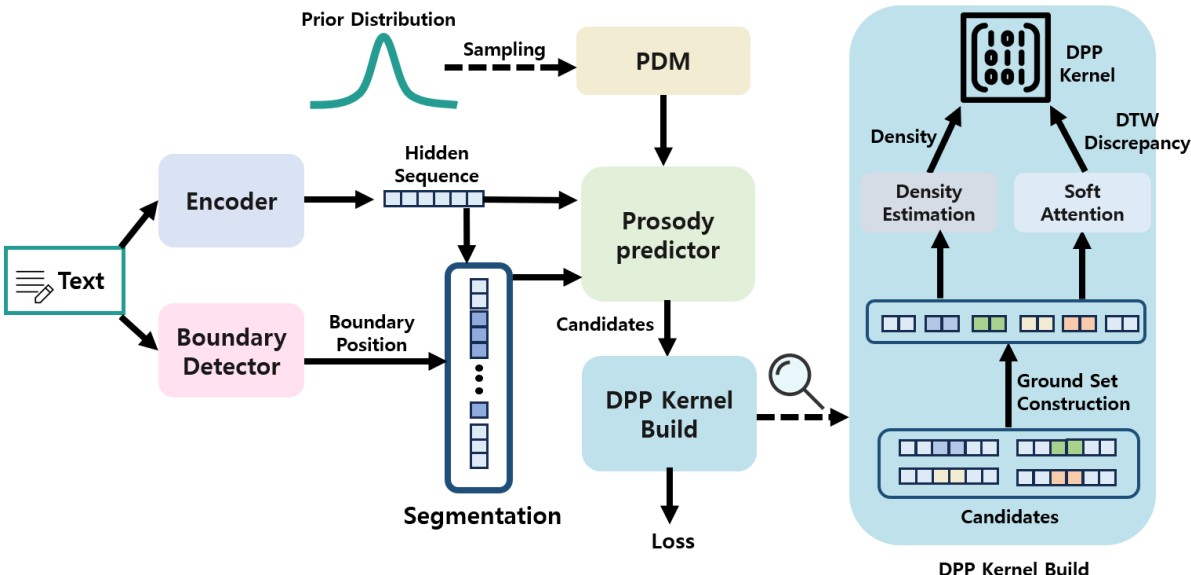

Figure 2: Diagrams describing the training procedure of PDM.

conditional DPPs, for a subset $B \subseteq Y$ not intersecting with $A$ we have

$$\mathcal{P}(Y = A \cup B | A \subseteq Y) = \frac{\mathcal{P}(Y = A \cup B)}{\mathcal{P}(A \subseteq Y)} \quad (2)$$

$$= \frac{\det(\boldsymbol{L}_{A \cup B})}{\det(\boldsymbol{L} + \boldsymbol{I}_{\bar{A}})}, \quad (3)$$

where $I_{\bar{A}}$ is the matrix with ones in the diagonal entries indexed by elements of $\mathcal{Y} - A$ and zeros elsewhere. In conditional DPPs, items in the ground set $\mathcal{Y}$ can be sampled according to kernel considering given contexts $A$. In this work, prosodic features corresponding to neighbor segments of the target are used as conditions for conditional DPPs.

## 2.3 Prosody phrasing

As humans usually speak out with some pauses to convey a message and the speaker's intention, the utterance is required to be divided into more fine-grained segments, referred as prosodic units. Ye et al. (2023) show that the prosody pattern reflecting relevant text context contributes to the enhanced TTS. Suni et al. (2020); Nguyen et al. (2020) suggest that prosodic boundary plays an important role in the naturalness and intelligibility of speech. To incorporate an inherent prosodic structure of input utterance within its context, we build Prosodic Boundary Detector (PBD) trained on a large corpus along with prominence labels.

## 3 DPP-TTS

Our model DPP-TTS is composed of base TTS based on FastSpeech2 (Ren et al., 2019), Prosody Diversifying module (PDM), and Prosody boundary Detector (PBD). Once the base TTS is trained, PDM is inserted in front of the prosody predictor and trained with the method which will be described in detail in Section 3.2. We describe the main modules of DPP-TTS in the following subsection 3.1. For the detailed architecture of DPP-TTS, refer to Appendix A.

### 3.1 Main modules of DPP-TTS

**Prosody predictor** Our prosody predictor estimates the distribution of the duration and pitch in prosody. For more human-like rhythm and pitch in prosody, the stochastic prosody predictor is built upon normalizing flows. Specifically, the stochastic duration predictor estimates the distribution of phoneme duration and the stochastic pitch predictor estimates the distribution of phoneme-level pitch from the hidden sequence. At the training stage, the prosody predictor learns the mapping from the distribution of prosodic features to normal distribution. At inference, it predicts the phoneme-level duration or pitch by reversing the learned flows. In addition, it also serves as the density estimator for prosodic features during the training of PDM which will be described in detail in the next subsection 3.2. The prosody predictor is trained to max-

imize a variational lower bound of the likelihood of the phoneme duration or pitch. More details regarding the prosody predictor are in Appendix A.

**PDM** Since the prosodic predictor is only trained with the lower bound of likelihood objective, it is not enough to generate diverse and non-monotonous prosody patterns. For more expressive speech modeling, PDM is added in front of the prosody predictor as shown in Figure 2. Its role is to map samples from a standard normal distribution to another distribution for diverse prosodic features of speech. This module is trained with an objective based on conditional DPPs which is described in Section 2.2. At inference of DPP-TTS, multiple prosodic candidates are generated by PDM. Subsequently, the prosodic feature of speech is selected via MAP inference when a single sample is generated, otherwise, multiple prosodic features are generated via the DPP sampling.

**Prosodic Boundary Detector** We utilize the prosodic boundary based on the prominence of words, by reflecting the assumption that people unconsciously pronounce the sentence focusing on what intentions they want to convey. To decide the boundary, we create Prosodic Boundary Detector (PBD) whose backbone is a pretrained Sentence-Transformer. A Prominence dataset (Talman et al., 2019), consisting of Librispeech scripts and prominence classes, is used for training our PBD. The input of PBD is a text sequence, and PBD predicts each word's prominence level. Based on the prominence level, the prosodic units are extracted from the input utterance.

## 3.2 Training process of PDM

In this section, we explain the methodology described in Algorithm 1 for training the prosody diversifying module (PDM). The training process mainly consists of three steps: segmentation of an input text, generation of prosodic feature candidates, and building DPP kernel. We also explain the conditional maximum induced cardinality (MIC) objective for training PDM. The overall training procedure is depicted in Figure 2.

### 3.2.1 Segmentation of the input text

In this stage, targets in input text for diversification of prosody are chosen. Given a sentence, PBD predicts the positions of prominent words. After that, those positions are used to divide the context and target sequences. Specifically, each target starts

at the prominent word and ends right before another prominent word. In addition, adjacent left and right contexts with the same number of words for the target are chosen.

---

**Algorithm 1** Training of PDM

---

**Require:** TextEncoder $f(\cdot)$, PDM parameterized $\theta$, a prosody predictor $g(\cdot)$, number of candidates $n_c$, noise scale $\epsilon$

1: **while** not converged **do**
2:    $h_{text} \leftarrow f(text)$
3:    Split $h_{text}$ into $[h_{target}, h_{context}]$ and store indices $\{i_t, i_c\}$ for the target and context
4:    Sample latent code $z_{context} \in \mathbb{R}^T$ with noise scale $\epsilon$
5:    Get prosodic features of contexts: $d_{context} \leftarrow [g(h_{text}, z_{context})]_{i_c}$
6:    Sample latent codes $z_{target}$ with noise scale $\epsilon$
7:    Get latent codes after PDM: $z_{target} \leftarrow \text{PDM}(z_{target}) \in \mathbb{R}^{n_c \times T}$
8:    Get $n_c$ prosodic features of targets: $(d_{target}^1, ...d_{target}^{n_c}) = [g(h_{text}, z_{target})]_{i_t}$
9:    Concatenate contexts and targets: $[d_{context}, d_{target}^i]$
10:   Get quality of targets along with contexts: $q_{target}^i = \texttt{Likelihood}([d_{context}, d_{target}^i])$
11:   Build the kernel of conditional DPPs: $L \leftarrow \text{Build kernel}(q_{target}^i, [d_{Con}, d_{target}^i])$
12:   Calculate the loss function: $L_{diversity} \leftarrow -\text{tr}(I - [(L + I_{\bar{A}})^{-1}]_{\bar{A}})$
13:   Update $\theta$ with the gradient $\nabla L_{diversity}$
14: **end while**

---

### 3.2.2 Generation of prosodic feature candidates

In this stage, multiple prosodic candidates are generated for DPP sampling as shown in Figure 2. First, a hidden sequence is generated from an input text through the text encoder. Second, the pretrained prosody predictor generates $n_c$ prosodic features conditioned on the hidden sequence by utilizing samples from the normal distribution. Meanwhile, other new samples from a normal distribution are fed into PDM, and then the prosodic predictor conditioned on the hidden sequence generates new prosodic features of the target from the output features of PDM. Finally, the latter-generated target prosodic features substitute former-generated target features, and then $n_c$ prosodic candidates are generated with the target and context entangled.

### 3.2.3 Construction of DPP kernel

Generated candidates are split into left and right context $d_L, d_R$ and $n$ targets $d_1, d_2, ...d_n$, then the candidate set for DPP is constructed as shown in the Figure 2. Next, the kernel of conditional DPP is built by incorporating both the diversity and quality(likelihood) of each candidate feature. Here, quality features guarantee the smooth transition of prosody. The kernel of conditional DPPs is defined as $L = \mathrm{diag}(q) \cdot S \cdot \mathrm{diag}(q)$, where $S$ is the similarity matrix and $q$ is the quality vector.

### 3.3 Objective function

**Similarity metric**  In the process of constructing the DPP kernel, we need to define the similarity metric between features. However, target sequences and context sequences often vary in length. As the Euclidean distance is not applicable to calculate the similarity between two sequences, we utilize Soft DTW:

$$S_{i,j} = \exp(-\mathbf{dtw}_\gamma^D(d_i, d_j)) \qquad (4)$$

, where $\mathbf{dtw}_\gamma^D$ denotes soft-DTW discrepancy with a metric $D$ and smoothing parameter $\gamma$. When the metric $D$ is chosen as the $L_1$ distance $D(x, y) = \sum_i |x_i - y_i|$ or half gaussian $D(x, y) = \|x - y\|_2^2 + \log(2 - \exp(-\|x - y\|_2^2))$, the similarity matrix becomes positive semi-definite (Blondel et al., 2021; Cuturi et al., 2007). In this work, $L_1$ distance is used as the metric of Soft-DTW so that $S$ to be positive semi-definite.

**Quality metric**  To reflect the naturalness of prosodic features, quality scores are calculated based on the estimated density of predicted features. Given the features $x$, posterior $q(z_i|x; \phi)$ and joint likelihood $p(x, z_i; \theta)$ where $\phi, \theta$ are parameters of the prosody predictor based on the variational method, the density values of predicted features are calculated with importance sampling using the prosody predictor: $p(x; \theta, \phi) \approx \sum_{i=1}^N \frac{p(x, z_i; \theta)}{q(z_i|x; \phi)}$. In the experiment, we empirically find that it is more helpful not to give a penalty to the quality score if the likelihood is greater than the specific threshold. With log-likelihood $\pi(x) = \log p(x)$, the quality score of single sample is defined as

$$q(x) = \begin{cases} w & \text{if } \pi(x) >= k \\ w \cdot \exp(\pi(x) - k) & \text{otherwise} \end{cases} \qquad (5)$$

, where $w$ is a quality weight and the threshold value $k$ was set as the average density of the training dataset in the experiment. We need to measure the diversity with respect to kernel $L$ to train the PDM. One straightforward choice is the maximum likelihood (MLE) objective, $\log \mathcal{P}_L(Y) = \log \det(L_Y) - \log \det(L + I)$. However, there are some cases where almost identical prosodic features are predicted. We recognize that such cases cause the objective value to become near zero, only to make the training process unstable. Instead, maximum induced cardinality (MIC) (Gillenwater et al., 2018) objective which is defined as $\mathbb{E}_{Y \sim \mathcal{P}_L}[|Y|]$ can be an alternative. It does not suffer from training instability. In this work, context segments $d_L, d_R$ are used as the condition in the MIC objective of conditional DPPs. The objective function with respect to the candidate set $[d_L, d_R, d_1, d_2, ..., d_N]$ and its derivative are as follows:

**Proposition 1 (MIC objective of CDPPs)** *With respect to the candidate set $[d_L, d_R, d_1, d_2, ..., d_N]$, the MIC objective of conditional DPPs and its derivative are as follows:*

$$L_{MIC} = \mathrm{tr}(I - [(L(\theta) + I_{\bar{A}})^{-1}]_{\bar{A}}), \qquad (6)$$

$$\frac{\partial L_{MIC}}{\partial \theta} = ((L + I_{\bar{A}})^{-1} I_{\bar{A}} (L + I_{\bar{A}})^{-1})^{\mathrm{T}} \frac{\partial L}{\partial \theta}, \qquad (7)$$

*where $A$ denotes the set of contexts $(d_L, d_R)$, $\bar{A}$ denotes the complement of the set $A$ and $\theta$ is the parameter of PDM.*

**Remark 1** *The MLE objective becomes unstable since the determinant volume is close to zero if two similar items are included. In contrast, the MIC objective guarantees stability as the gradient of our objective guarantees the full-rank structure.*

Detailed proof is presented in Appendix B. At inference, for predicting a single prosodic feature, MAP inference is performed across sets with just a single item as follows: $x^* = \arg\max_{x \in \bar{A}} \log \det(L_{\{x\} \cup A})$. Otherwise, the k-DPP sampling method is used for sampling multiple prosodic features when multiple speech samples are generated. The detailed procedure of training PDM and the inference of DPP-TTS are in Appendix C.

## 4 Experiment setup

### 4.1 Dataset and preprocessing

We conduct experiments on the LJSpeech dataset which consists of audio clips with approximately

24 hours lengths. We split audio samples into 12500/100/500 samples for the training, validation, and test set. Audio samples with 22kHz sampling rate are transformed into 80 bands mel-spectrograms through the Short-time Fourier transform (STFT) with 1024 window size and 256 hop length. International Phonetic Alphabet (IPA) sequences are used as input for phoneme encoder. Text sequences are converted to IPA phoneme sequences using `Phonemizer`[1] software. Following (Kim et al., 2020), the converted sequences are interspersed with blank tokens, which represent the transition from one phoneme to another phoneme.

## 4.2 Implementation Details

Both the base TTS and PDM are trained using the AdamW optimizer (Loshchilov and Hutter, 2019) with $\beta_1 = 0.8, \beta_2 = 0.99$ and $\lambda = 0.01$. The initial learning rate is $2 \times 10^{-4}$ for the basic TTS and $1 \times 10^{-5}$ for the PDM with exponential learning decay. Base TTS is trained with 128 batch size for 270k steps on 4 NVIDIA RTX A5000 GPUs. PDM is trained with 8 batch size for 2k steps on single GPU. The quality weight of the model is set as $w = 10$. We prepare two DPP-TTS versions for the evaluation: (1): **DPP-TTS-d**: a model that has the duration diversifying module and **DPP-TTS-p**: a model that has the pitch diversifying module.

## 4.3 Baselines

We compare our model with the following state-of-the-art models: 1) VITS[2] (Kim et al., 2021), an end-to-end TTS model based on conditional VAE and normalizing flows. It mainly has two parameters for the sampling: the standard deviation of input noise $\sigma$ to duration predictor and a scale factor $\tau$ to the standard deviation of prior distribution which controls other variations of speech (e.g., pitch and energy); 2) Flowtron[3] (Valle et al., 2021), an autoregressive flow-based TTS model; 3) DiffSpeech[4] (Liu et al., 2021), a diffusion-based probabilistic text-to-speech model; 4) SyntaSpeech[5] (Ye et al., 2022), a TTS model using a syntactic graph of input sentence for prosody modeling. HiFi-GAN (Kong et al., 2020) is used as the vocoder for synthesizing waveforms from

the mel-spectrograms. In addition, we also compare our model with baseline DPP-TTS w/o PDM which does not utilize PDM for prosody modeling. Audio samples used for the evaluation are in the supplementary material and the Demo page[6].

## 4.4 Evaluation Method

**Side-by-Side evaluation** To evaluate the perceptual diversity of our model, we conduct a side-by-side evaluation of the prosody of speech samples. Via Amazon Mechanical Turk (AMT), we assign ten testers living in the United States to a pair of audio samples (i.e., DPP-TTS and a baseline), and ask them to listen to audio samples and choose among three options: **A**: sample A has more varied prosody than sample B, **Same**: sample A and B are equally varied in prosody, **B**: the opposite of first option. For DPP-TTS-d, testers are asked to focus on the **rhythmic** variation of the speech sample other than different aspects of prosody. Likewise for DPP-TTS-p, the testers are asked to focus on the **pitch** variation of speech samples. Importantly, the testers are asked to ignore about pace, and volume of speech samples to eliminate possible bias on speech samples as possible.

**MOS** In addition, we also conduct the Mean-Opinion-Score (MOS) to evaluate the naturalness of prosody for generated samples. Ten testers are assigned to each audio sample. Given reference speech samples to each score, testers are asked to give a score between 1 to 5 on a 9-scale based on the sample's naturalness, In addition, they are asked to focus on the prosody aspect of audio samples.

**Quantitative evaluation** In addition to human evaluation, We conduct quantitative evaluations for our DPP-TTS, DPP-TTS w/o PDM, and VITS with high temperature [7]. We have used the following metrics for evaluating our model and the baseline.

$\sigma_p$: phoneme-level standard deviation of duration or pitch in a speech. This metric reflects the prosodic diversity inside each speech sample.

**Determinant**: a determinant of the similarity matrix is used to evaluate the diversity among prosodic features of 10 generated samples. Cosine similarity is used as a metric between two features. Since dissimilar items increase the volume of the matrix, higher determinant values indicate that generated samples are more diverse.

---

[1] https://github.com/bootphon/phonemizer
[2] https://github.com/jaywalnut310/vits
[3] https://github.com/NVIDIA/flowtron
[4] https://github.com/MoonInTheRiver/DiffSinger
[5] https://github.com/yerfor/SyntaSpeech

[6] https://dpp-tts.github.io/
[7] Same counterparts as the side-by-side evaluation are used.

Table 1: Side-by-side comparison on the LJSpeech dataset along with MOS results. Model A/B denotes the models compared for the side-by-side evaluation. 'A' denotes the percentage of testers who vote for the sample of Model A is more varied than B, and 'B' denotes the vice-versa. 'Same' denotes the percentage of testers who vote for two samples that are equally varied in prosody.

| Model A/B | A | Same | B | Model A MOS | Model B MOS |
|---|---|---|---|---|---|
| DPP-TTS-d/Flowtron($\tau = 0.6$) | 52.0% | 16.5% | 31.5% | $3.97 \pm 0.08$ | $3.85 \pm 0.06$ |
| DPP-TTS-p/Flowtron($\tau = 1.0$) | 49.3% | 22.6% | 28% | $4.02 \pm 0.08$ | $3.66 \pm 0.09$ |
| DPP-TTS-p/VITS($\tau = 0.667, \sigma = 0.8$) | 57.25% | 6.75% | 36% | $4.02 \pm 0.08$ | $4.38 \pm 0.07$ |
| DPP-TTS-d/VITS($\tau = 0.667, \sigma = 1.2$) | 44% | 34.6% | 21.3% | $3.97 \pm 0.08$ | $3.95 \pm 0.08$ |
| DPP-TTS-p/VITS($\tau = 1.2, \sigma = 0.8$) | 34.6% | 42.3% | 23% | $4.02 \pm 0.08$ | $3.98 \pm 0.08$ |
| DPP-TTS-d/DiffSpeech | 55.5% | 12.7% | 31.8 % | $3.97 \pm 0.08$ | $3.92 \pm 0.06$ |
| DPP-TTS-p/DiffSpeech | 66.4% | 5.4 % | 28.2 % | $4.02 \pm 0.08$ | $3.92 \pm 0.06$ |
| DPP-TTS-d/SyntaSpeech | 55.5% | 9.1% | 35.4 % | $3.97 \pm 0.08$ | $4.04 \pm 0.09$ |
| DPP-TTS-p/SyntaSpeech | 63.6% | 6.4 % | 30.0 % | $4.02 \pm 0.08$ | $4.04 \pm 0.09$ |
| **DPP-TTS-p/DPP-TTS w/o PDM** | 55.5 % | 17.5 % | 27 % | $4.02 \pm 0.08$ | $4.09 \pm 0.09$ |

**Inference time**: the inference time for synthesizing a waveform is calculated in the TTS model. The inference speed is evaluated on Intel(R) Core(TM) i7-7800X CPU and a single NVIDIA RTX 3080 GPU. Computation time is averaged over 100 forward passes.

## 5 Results

### 5.1 Perceptual Diversity Results

We first report the side-by-side evaluation results between our model and baselines along with the MOS test. Results are shown in Table 1. We observe three findings from the evaluation: 1) Scaling temperature of Flowtron and VITS model does not contribute much to the perceptual diversity of rhythm and pitch while the MOS degrades by a large margin; 2) Our DPP-TTS model outperforms the DPP-TTS w/o PDM in the side-by-side evaluation which proves the usefulness PDM with MIC training objective; 3) Our model DPP-TTS outperforms the four baselines in the side-by-side evaluation for rhythm and pitch comparison. Since LJSpeech dataset consists of short text transcripts, we conduct the side-by-side evaluation for paragraph samples to evaluate prosody modeling in longer input texts. As the evaluation of the LJSpeech dataset, listeners are asked to choose among the three options. In the side-by-side evaluation of prosody in a paragraph, both the DPP-TTS-d/p outperform the baseline as Table 2 shows. More testers voted for our model in the evaluation for paragraph than the LJSpeech dataset. We speculate that this is due to diverse prosody patterns being more prominent in longer texts.

Table 2: Side-by-Side comparison on the paragraph.

| Model A/B | A | Same | B |
|---|---|---|---|
| DPP-TTS-p/VITS($\tau = 0.667, \sigma = 0.8$) | 61.0% | 13.75% | 25.25% |
| DPP-TTS-d/VITS($\tau = 0.667, \sigma = 1.2$) | 48.45% | 25.27% | 26.27% |
| DPP-TTS-p/VITS($\tau = 1.2, \sigma = 0.8$) | 40.91% | 36.34% | 22.73% |

### 5.2 Quantitative evaluation

Table 3: Quantitative evaluation results. Numbers in bold denote the better result.

| Model | $\sigma_p$ | Duration Determinant |
|---|---|---|
| VITS | 0.039 | $1.86 \times 10^{-5}$ |
| DPP-TTS-d | **0.041** | $\mathbf{2.98 \times 10^{-5}}$ |
| DPP-TTS w/o PDM | 0.033 | $2.02 \times 10^{-5}$ |

| Model | $\sigma_p$ | Pitch Determinant | Inference time(s) |
|---|---|---|---|
| Flowtron | 0.0121 | $6.76 \times 10^{-14}$ | $5.3 \times 10^{-1}$ |
| VITS | 0.0132 | $4.12 \times 10^{-15}$ | $\mathbf{1.3 \times 10^{-2}}$ |
| DiffSpeech | 0.0133 | $8.51 \times 10^{-14}$ | $4.1 \times 10^{-2}$ |
| SyntaSpeech | 0.0145 | $2.23 \times 10^{-14}$ | $3.9 \times 10^{-2}$ |
| DPP-TTS-p | **0.0145** | $\mathbf{1.88 \times 10^{-13}}$ | $4.4 \times 10^{-2}$ |
| DPP-TTS w/o PDM | 0.011 | $3.13 \times 10^{-14}$ | $2.3 \times 10^{-2}$ |

Table 3 shows the result of $\sigma_p$, determinant, and inference time[8]. In both duration and pitch $\sigma_p$, DPP-TTS outperforms baselines by having a higher standard deviation in the phoneme-level features. It demonstrates that DPP-TTS generates a speech with more dynamic pitch and rhythm than the baseline. The determinants of duration and pitch sets of

---

[8]If the model does not have a separate duration predictor like VITS, it is challenging to evaluate the duration of phonemes, therefore only the pitch evaluation is included for other baselines.

Table 4: Side-by-side comparison between the model with PBD and the fixed-length baseline without PBD.

| Model A/B | A | Same | B | Model A MOS | Model B MOS |
|---|---|---|---|---|---|
| DPP-TTS-d/Fixed-length | 58.2% | 24.5% | 17.3% | $4.02 \pm 0.08$ | $3.78 \pm 0.06$ |
| DPP-TTS-p/Fixed-length | 61.8% | 23.4% | 14.8% | $3.98 \pm 0.08$ | $3.78 \pm 0.06$ |

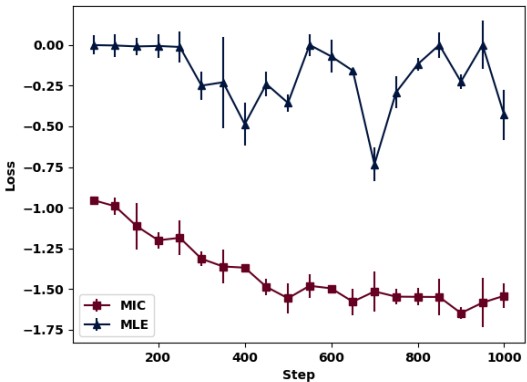

Figure 3: Loss trajectory for the MIC and MLE objective. While the training dynamic of MLE objective is unstable due to the structure, the MIC objective shows faster convergence.

DPP-TTS also outperform the baseline. It shows that DPP-TTS generates more samples with diverse prosody than baselines. Finally, DPP-TTS results in 0.044 seconds of inference speed. Although its inference speed is slower than the baseline, our model is applicable in practice since the inference speed of our model is 22.7x faster than real-time.

### 5.3 Model analysis

**MIC vs MLE objective for training PDM**   We conduct ablation studies to verify the effectiveness of MIC training objective for conditional DPPs. Fig 3 shows the training loss trajectory. The norm of the gradient for MLE objective blows up when prosodic features in the candidate set are nearly identical leading to unstable training. In contrast, since our MIC objective guarantees the full rank structure of the gradient matrix, the training does not suffer from instability, leading to faster convergence.

**Adjusting the extent of variation**   To study the impact of quality weight $w$ in quality metric $\boldsymbol{q}(x)$ for DPP sampling, we generate multiple speech samples with different values of quality weight. For a large magnitude of quality weight, our model likely generates smoother prosody patterns. As the magnitude of quality weight $w$ decreases, the

model starts to generate more dynamic and diverse prosody patterns. For examples of the pitch contour from generated samples, refer to Appendix G.

**Case study**   To analyze why and how the naturalness drops after applying PDM, we investigate some samples that get low MOS by testers. The degradation cases mainly fall into two cases. First, in some utterances, the prosodic transition between adjacent words arises too rapidly. A quality metric that imposes a penalty for the rapid transition may address this problem. Second, speech realization at a particular level of pitch is somewhat awkward. This problem is likely to appear since the TTS model has not seen enough prosody patterns during the training stage. We believe that speech augmentation related to pitch or duration will address this problem.

**Effectiveness of PBD**   We conduct an experiment to verify the effectiveness of PBD over the fixed-length baseline. While PBD dynamically adjusts the length of the target considering the prosodic boundary, the fixed-length baseline just extracts the target of fixed length. For example, with length $n = 3$, the first three words are selected as the context, the next three words are selected as the target, and the next three words are selected as the context, and so on. We can see from Table 4, both the perceptual diversity and MOS of the baseline decrease compared to the model using PBD. It indicates the advantage of PBD which dynamically adjusts the target length considering the prosodic boundary.

## 6   Related Works

There have been many efforts in TTS research to enhance the expressivity of generated speech. Learning latent prosody embedding at the sentence level is proposed to generate more expressive speech (Skerry-Ryan et al., 2018; Wang et al., 2018). Wan et al. (2019) presents a hierarchical conditional VAE model to generate speech with more expressive prosody and Sun et al. (2020) propose a hierarchical VAE-based model for fine-grained prosody modeling. In addition, a speech syn-

thesis model incorporating linguistic information BERT (Devlin et al., 2019) is proposed to get enriched text representation by (Kenter et al., 2020). However, the controllability of speech attributes like pitch and rhythm is not fully resolved and the expressivity or diversity of generated speech is still far off the human.

Meanwhile, since prosody is directly related to duration, pitch, and energy, text-to-speech models that explicitly control these features are proposed (Lańcucki, 2021; Ren et al., 2021a). A flow-based stochastic duration predictor is also proposed to generate speech with more diverse rhythms and it shows superior performance compared to a deterministic duration predictor (Kim et al., 2021). In this work, we incorporate a stochastic duration, and pitch predictor upon Fastspeech2 (Ren et al., 2021a) to model more expressive prosody.

## 7 Conclusion

We propose a Prosody Diversifying Module (PDM) that explicitly diversifies prosodic features like duration or pitch to avoid a monotonous speaking style, using conditional MIC objective. Previous research has indicated that well-positioned prosodic boundaries are helpful for understanding the meaning of speech (Sanderman and Collier, 1997). There are some possible avenues to improve our model for more realistic prosody. A more sophisticated segmentation rule that reflects human prosodic boundaries can improve the naturalness of our model. Incorporating linguistic representation well-aligned with human intonation contributes to accurate prosodic boundaries. Such methods will boost the performance of our model.

## Limitations

Since our methodology depends on the segmentation of input utterances, a sophisticated boundary for input utterances is necessary. We believe that more advanced research for prosody boundary detectors will contribute to smoother and more expressive prosody for TTS. The expression of speech at a certain pitch level can be somewhat unnatural. This issue probably arises because the TTS model hasn't been sufficiently exposed to prosody patterns during its training phase. We believe that improvements related to pitch or duration in speech augmentation could solve this issue. We plan to resolve this problem by using an advanced augmentation method and large pretrained models. For the

quality metric for building DPP kernel, we have used the density value of each sample. We use an importance sample scheme for the value estimation, however, it needs multiple samples for exact evaluation. We believe a more efficient sampling scheme contributes to generating a more exact evaluation of the naturalness of prosodic features.

## Acknowledgements

We thank anonymous reviewers for their constructive and insightful comments. K. Jung is with ASRI, Seoul National University, Korea. This work was supported by Samsung Electronics. This work was partly supported by Institute of Information communications Technology Planning Evaluation (IITP) grant funded by the Korea government(MSIT) [NO.2021-0-01343, Artificial Intelligence Graduate School Program (Seoul National University)].

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

# A  Details of DPP-TTS

Our model DPP-TTS is composed of a Seq2Seq module for generating the mel-spectrogram, a prosody predictor for predicting duration and pitch sequences, and a prosody diversifying module (PDM). At the first stage, the base TTS model which consists of the Seq2Seq module and the prosody predictor is trained as shown in Figure 4a. Once the base TTS is trained, PDM is inserted in front of the prosody predictor and trained with the method which will be described in detail in Section 3.2. We describe the main modules of DPP-TTS and their roles in the following subsection.

## A.1  Main modules of DPP-TTS

**Seq2Seq module**  The role of the Seq2seq module is generating mel-spectrograms from phoneme sequences. The module is adapted from Fast-Speech2 (Ren et al., 2021a) with some modifications. The model consists of four main parts: a phoneme prenet, a phoneme encoder, a variance adaptor, and a mel-spectrogram decoder. In the phoneme encoder, phoneme sequences are processed through a stack of feed-forward transformer blocks with a relative positional representation (Shaw et al., 2018). In variance adaptor at training, pitch embeddings and energy embeddings [9] are added to encoded hidden representations and then hidden representations are expanded according to ground-truth duration labels [10]. At inference, these prosodic features are provided from predictions of the prosody predictor. Finally, expanded representations are processed through a stack of feed-forward transformer blocks and mel-spectrograms with 80 channels are generated after the linear projection. The Seq2Seq module is trained to minimize $L_1$ distance between the predicted and target mel-spectrogram.

**Prosody predictor**  In FastSpeech2, the variance adaptor consists of deterministic predictors for predicting prosodic features. However, a deterministic prosodic predictor is not expressive enough to learn the speaking style of a person. For diverse rhythm and pitch, a stochastic duration predictor and pitch predictor are built upon normalizing flows. Specifically, the stochastic duration predictor estimates the distribution of phoneme duration and the stochastic pitch predictor estimates the distribution of phoneme-level pitch from the hidden sequence. At the training stage, the prosody predictor learns the mapping from the distribution of prosodic features to normal distribution. At inference, it predicts the phoneme-level duration or pitch by reversing the learned flows. In addition, it also serves as the density estimator for prosodic features during the training of PDM which will be described in detail in Section 4. The prosody predictor is trained to maximize a variational lower bound of the likelihood of the phoneme duration or pitch. More details regarding the prosody predictor are in Appendix A.

**PDM**  Although the stochastic duration and pitch predictor are trained to generate a speech with diverse rhythm and pitch, the prosody predictor may favors major modes and it can lead to the monotonous prosodic pattern in the speech. For more expressive speech modeling, PDM is added in front of the prosody predictor as shown in Figure 4b. Its role is to map latent codes from a standard normal distribution to another distribution for diverse prosodic features of speech. This module is trained with an objective based on conditional DPPs which is described in Section 2.2. At inference of DPP-TTS, multiple prosodic candidates are generated by PDM. Subsequently, the prosodic feature of speech is selected via MAP inference, or multiple prosodic features are sampled when multiple speech samples are generated.

**Prosodic Boundary Detector**  There are two ways of diversifying prosodic features: autoregressive and non-autoregressive methods. Since autoregressive methods require much time to generate the output sequence, we employ the non-autoregressive method in our model. For the non-autoregressive approach, some prosodic features of speech need to be set when trying to diversify overall prosodic features. In this perspective, a sentence needs to be divided into a context sequence and target sequence, utilizing the prosodic boundary based on the prominence of words by reflecting the assumption that people unconsciously pronounce the sentence focusing on what intentions they want to convey. To decide the boundary, we make Prosodic Boundary Detector (PBD) whose backbone is a pretrained Sentence-Transformer. A Prominence dataset (Talman et al., 2019) consisting of Librispeech scripts

---

[9]For the brevity, the deterministic energy predictor is omitted in the figure.

[10]Ground-truth labels are obtained via monotonic alignment search (Kim et al., 2020) between the phonemes and mel-spectrogram.

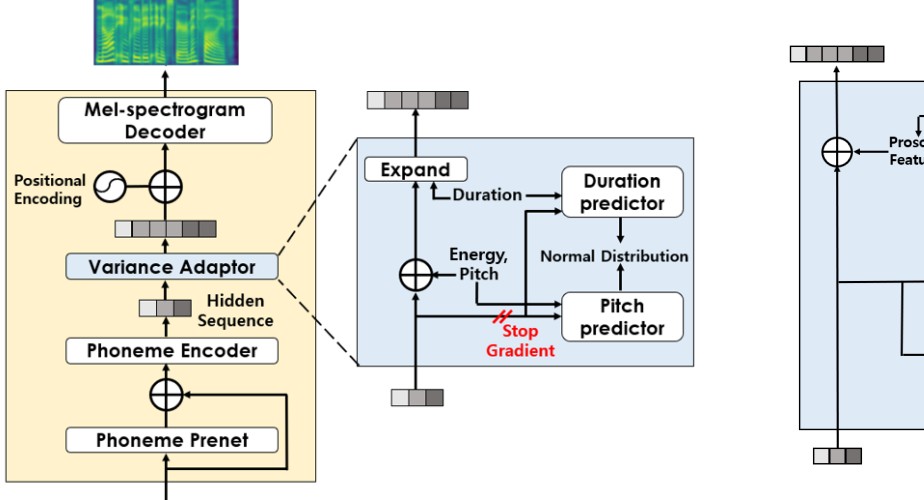

(a) Base text-to-speech model training

(b) Variance adaptor at inference of DPP-TTS

Figure 4: Diagrams describing (a): training of base text-to-speech model and (b) variance adaptor at the inference of DPP-TTS after PDM is added.

and prominence classes is used for training PBD. The input of PBD is a text sequence, and PBD predicts each word's prominence level.

## B  Details of prosody predictor and PDM

### B.1  Prosody predictor

**Training**  It is hard to use the maximum likelihood objective directly to train duration predictor because the duration of each phoneme is 1) a discrete integer and a scalar, which hinders expressive transformation because invertibility should remain in normalizing flow. To train duration predictor, duration values are extended to continuous values using variational dequantization (Ho et al., 2019) and are augmented with extra dimensions using variational data augmentation (Chen et al., 2020). Specifically, duration sequence $d$ becomes continuous as $d - u$ where $u$'s value is restricted to $[0, 1)$ and augmented as $[d - u, v]$ with a extra random variable $v$. Two random variables $u$ and $v$ are sampled through approximate posterior $q_\phi(u, v|d, h_{text})$. The ELBO can be calculated as follows:

$$\log p_\theta(d|h_{text}) \geq$$
$$\mathbb{E}_{q_\phi(u,v|d,h_{text})}[\log \frac{p_\theta(d - u, v|h_{text})}{q_\phi(u, v|d, h_{text})}]$$

Like the duration predictor, the ELBO for pitch predictor can be calculated as follows:

$$\log p_\theta(p|h_{text}) \geq$$
$$\mathbb{E}_{q_\phi(v|p,h_{text})}[\log \frac{p_\theta(p, v|h_{text})}{q_\phi(v|p, h_{text})}],$$

where $p$ denotes the pitch sequences. Both the duration and pitch predictor are trained on negative lower bound of likelihood.

**Architecture**  The pitch predictor and duration predictor share identical architecture except for the additional random variable $u$. We will introduce the architecture of the duration predictor whose diagram is shown in Figure 5a. Duration predictor consists of condition encoder for hidden sequence, posterior encoder for the duration sequence, and the main flow blocks $g(\cdot)$. Specifically, posterior encoder maps latent codes from normal distribution to random variable $[u, v]$ and flow $g(\cdot)$ maps $[d - u, v]$ to normal distribution. Figure 5b shows the coupling of normalizing flows. First, input $x$ is split into $[x_0, x_1]$ and then $x_0$ is processed by a 1x1 convolution block. The output of the convolution block is processed by dilated convolution and Nystromer block and their outputs are concatenated. The concatenated output is processed by LayerNorm, Mish activation, and Convolution block. Finally, spline flows (Durkan et al., 2019) are parameterized by the output and then the output of spline flows $y$ and $x_0$ are concatenated.

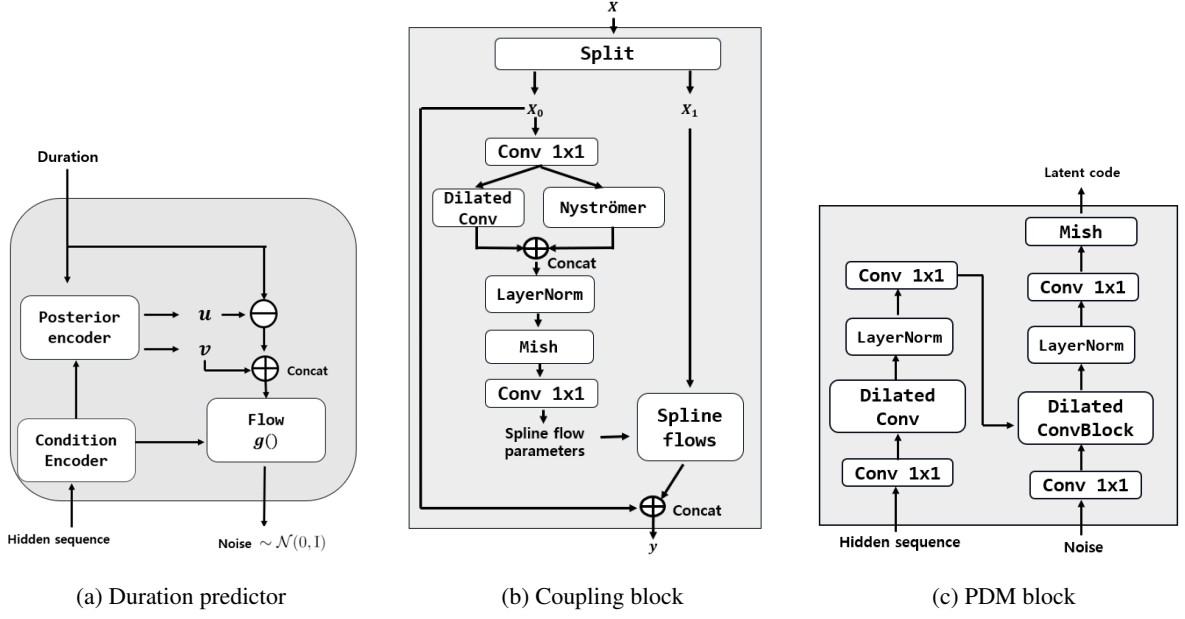

(a) Duration predictor       (b) Coupling block       (c) PDM block

Figure 5: Diagrams describing (a): the duration predictor, (b): Coupling block in normalizng flows and (c): Prosody diversifying module.

## B.2 PDM

The architecture of PDM is shown in Figure 5c. First, the hidden sequence is processed by convolution block followed by the dilated convolution block conditioned on hidden sequences, Layernorm, and 1x1 convolution block. After that, noise from a normal distribution is processed by 1x1 convolution followed by dilated convolution block, LayerNorm, and Mish activation.

As the prosody pattern of human speech is correlated with the local context and global context of the text, we design the architecture of the model to encode both the local and global features. Specifically, dilated convolution blocks are stacked to encode local features and a transformer block is used to encode global features in the coupling layers of normalizing flows. However, using the vanilla transformer to encode the global features in the coupling layer requires too large computational complexity. Therefore, Nyströmformer (Xiong et al., 2021) which is a Nyström-Based algorithm for approximating self-attention is used to encode global context for more efficient memory usage in the coupling layer of normalizing flow. Encoded local features and global features are concatenated and processed through LayerNorm (Ba et al., 2016), Mish activation (Misra, 2019) and a 1x1 convolution module.

Following the duration predictor model in Kim et al. (2021), variational dequantization (Ho et al.,

2019) is used for the duration predictor since the phoneme duration is a discrete integer. In addition, variational augmentation (Chen et al., 2020) is used to expand channel dimensions for expressive flows in both the duration and pitch predictor. The stochastic and pitch predictor are trained by maximizing their variational lower bounds (ELBO). Details of the duration and pitch predictor are described in Appendix B.

## C Proof of Proposition 1

**Objective** From equation [45] in (Kulesza and Taskar, 2012), the marginal kernel of conditional DPPs given the appearance of set $A$ has a following form:

$$\boldsymbol{K}^A = \boldsymbol{I} - [(\boldsymbol{L} + \boldsymbol{I}_{\bar{A}})^{-1}]_{\bar{A}} \qquad (8)$$

In addition, from equation [34] in Kulesza and Taskar (2012) the expected cardinality of $\boldsymbol{Y}$ given the marginal kernel $\boldsymbol{K}$ is:

$$\mathbb{E}[|\boldsymbol{Y}|] = \sum_{n=1}^{N} \frac{\lambda_n}{\lambda_n + 1} = \mathrm{tr}(\boldsymbol{K}) \qquad (9)$$

From equation 8, 9, the expected cardinality of conditional DPP given the appearance of set $A$ is:

$$\mathbb{E}[|\boldsymbol{Y}|] = \mathrm{tr}(\boldsymbol{K}^A) = \mathrm{tr}(\boldsymbol{I} - [(\boldsymbol{L} + \boldsymbol{I}_{\bar{A}})^{-1}]_{\bar{A}}) \quad (10)$$

**Derivative** For the proof, we will start with the following lemma:

**Lemma 1** *Given a matrix E and non-singular matrix A, following equation holds:*

$$\frac{\partial}{\partial A}\mathrm{tr}(E^{\mathrm{T}}A^{-1}E) = -(A^{-1}EE^{\mathrm{T}}A^{-1})^{\mathrm{T}} \quad (11)$$

*Proof.* First, consider $\frac{\partial}{\partial A_{ij}}tr(E^{\mathrm{T}}A^{-1}E)$. Since *trace* and derivative operator are interchangeable,

$$\frac{\partial}{\partial A_{ij}}\mathrm{tr}(E^{\mathrm{T}}A^{-1}E) = \mathrm{tr}(\frac{\partial}{\partial A_{ij}}(E^{\mathrm{T}}A^{-1}E))$$
$$= -\mathrm{tr}(E^{\mathrm{T}}A^{-1}\frac{\partial A}{\partial A_{ij}}A^{-1}E)$$
$$(12)$$

By setting $\frac{\partial A}{\partial A_{ij}} = E^{ij}$ where $E^{ij}$ denotes the matrix whose $(i,j)$ component is 1 and 0 elsewhere and $\boldsymbol{C} = -E^{\mathrm{T}}A^{-1}E^{ij}A^{-1}E$,

$$-\mathrm{tr}(E^{\mathrm{T}}A^{-1}E^{ij}A^{-1}E) = \sum_{i'} C_{i'i'}$$
$$= -\sum_{i'}\sum_{k_1}\sum_{k_2}(E^{\mathrm{T}}A^{-1})_{i'k_1}E^{ij}_{k_1k_2}(A^{-1}E)_{k_2i'}$$
$$= -\sum_{i'}(E^{\mathrm{T}}A^{-1})_{i'i}(A^{-1}E)_{ji'} =$$
$$= -\sum_{i'}(A^{-\mathrm{T}}E)_{ii'}(E^{\mathrm{T}}A^{-\mathrm{T}})_{i'j}$$
$$= -(A^{-\mathrm{T}}EE^{\mathrm{T}}A^{-\mathrm{T}})_{ij}$$
$$= -(A^{-1}EE^{\mathrm{T}}A^{-1})^{\mathrm{T}}_{ij}$$
$$(13)$$
$$\implies \frac{\partial}{\partial A_{ij}}\mathrm{tr}(E^{\mathrm{T}}A^{-1}E) = -(A^{-1}EE^{\mathrm{T}}A^{-1})^{\mathrm{T}}_{ij}.$$

Now, with respect to set a $A$ whose cardinality is $p$ and matrix $\boldsymbol{L} \in \mathbb{R}^{(p+q)\times(p+q)}$

$$\frac{\partial}{\partial\theta}\mathrm{tr}(\boldsymbol{I} - [(\boldsymbol{L} + \boldsymbol{I}_{\bar{A}})^{-1}]_{\bar{A}})$$
$$= -\frac{\partial}{\partial\theta}\mathrm{tr}([(\boldsymbol{L} + \boldsymbol{I}_{\bar{A}})^{-1}]_{\bar{A}}) \quad (14)$$
$$= -\frac{\partial}{\partial\theta}\mathrm{tr}(E^{\mathrm{T}}(\boldsymbol{L} + \boldsymbol{I}_{\bar{A}})^{-1}E),$$

where $E$ denotes $\begin{bmatrix} \boldsymbol{0} \\ \boldsymbol{I}_q \end{bmatrix} \in \mathbb{R}^{(p+q)\times q}$. Then by Lemma 1.

$$-\frac{\partial}{\partial\theta}\mathrm{tr}(E^{\mathrm{T}}(\boldsymbol{L} + \boldsymbol{I}_{\bar{A}})^{-1}E)$$
$$= ((\boldsymbol{L} + I_{\bar{A}})^{-1}EE^{\mathrm{T}}(\boldsymbol{L} + \boldsymbol{I}_{\bar{A}})^{-1})^{\mathrm{T}}\boldsymbol{L}'(\theta) \quad (15)$$
$$= ((\boldsymbol{L} + I_{\bar{A}})^{-1}\boldsymbol{I}_q(\boldsymbol{L} + \boldsymbol{I}_{\bar{A}})^{-1})^{\mathrm{T}}\boldsymbol{L}'(\theta)$$

The proof of Proposition 1 is now finished.

## D   Inference of DPP-TTS

---
**Algorithm 2** Inference of DPP-TTS
---
**Require:** TextEncoder $f(\cdot)$, Decoder $h(\cdot)$, PDM, a prosody predictor $g(\cdot)$, noise scale $\epsilon$
1:  $h_{text} \leftarrow f(text)$
2:  Split $h_{text}$ into $[h_{target}, h_{context}]$
3:  Sample latent code $z_{context} \in \mathbb{R}^{rmT}$ with noise scale $\epsilon$
4:  Get prosodic features of contexts: $d_{context} \leftarrow g^{-1}(h_{context}, z_{context})$
5:  Get quality of contexts: $q_{context} = $ Density estimation$(d_{context})$
6:  Sample latent codes $z_{target}$ with noise scale $\epsilon$
7:  Get latent codes after PDM: $z_{target} \leftarrow$ PDM$(z_{target}) \in \mathbb{R}^{n_c \times T}$
8:  Get $n_c$ prosodic features of targets: $(d^1_{target}, d^2_{target}, ...d^{n_c}_{target})$
9:  Get quality of targets: $q_{target} = $ Density estimation$(d_{target})$
10:  Concatenate contexts and targets: $[q_{context}, q_{target}], [d_{context}, d_{target}]$
11:  Build the kernel of conditional DPPs: $\boldsymbol{L} \leftarrow$ Build kernel$([q_c, q_t], [d_c, d_t])$
12:  Perform the MAP inference: $d^* \leftarrow \arg\max_d \log\det(\boldsymbol{L}_{d \cup d_{context}})$
13:  Synthesize wavs with prosodic features: $y \leftarrow h(h_{text}, d^*)$
---

Table 5: Side-by-side comparison between models with different quality weight values. DPP-TTS with quality weight $w = 2.0$ is used as the Model A.

| Model A/B | A | Same | B | Model A MOS | Model B MOS |
|---|---|---|---|---|---|
| DPP-TTS/DPP-TTS-w=1.0 | 24.5% | 17.7% | 57.8% | $4.02 \pm 0.08$ | $3.95 \pm 0.06$ |
| DPP-TTS/DPP-TTS-w=5.0 | 44.5% | 24.2% | 31.8% | $4.02 \pm 0.08$ | $4.07 \pm 0.06$ |
| DPP-TTS/DPP-TTS-w=10.0 | 57.6% | 19.5% | 22.9% | $4.02 \pm 0.08$ | $4.11 \pm 0.06$ |

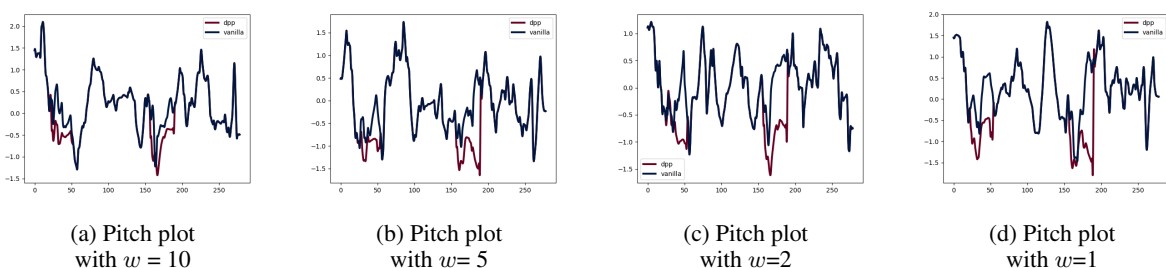

(a) Pitch plot with $w = 10$    (b) Pitch plot with $w= 5$    (c) Pitch plot with $w=2$    (d) Pitch plot with $w=1$

Figure 6: Pitch and log duration plots with different values of quality weight. Green plots indicate the prosody prediction before MAP inference of DPP and red plots indicate the prosody prediction after MAP inference of DPP.

## E  Sample paragraph for the side-by-side comparison test

```
  Known individually and collectively
as ShaiHulud, the sandworms are these
supermassive beings that plow through
the deserts of Arrakis, consuming
everything that dares venture unprepared
into their territory.  The worms are
what make harvesting spice so difficult
because they tend to eat whatever tools
off-worlders use to mine it.  They are
also sacred to the Fremen, who seem to
know ways to navigate around them, and,
somehow, they're linked to the creation
of spice.  Think of them as big honking
metaphors for the sublime powers of nature
that loom beyond human understanding,
like a desert full of Moby Dicks
```

## F  Adjusting the extent of variation

Table 5 demonstrates that adjusting quality weight can trade-off between diversity and naturalness without having a severe impact on naturalness. The pitch and log-duration plots are shown in Figure 6.