# OpenReview forum: "DPP-TTS: Diversifying prosodic features of speech via determinantal point processes"
_EMNLP/2023/Conference — EMNLP 2023 Main_

### Official Review · Reviewer_a3rZ · 2023-07-29

**Soundness:** 3

**Excitement:**

4: Strong: This paper deepens the understanding of some phenomenon or lowers the barriers to an existing research direction.

**Paper Topic And Main Contributions:**

This paper proposes to use the Determinantal Point Processes (DPP) for diverse prosody modelling in text-to-speech synthesis. It proposes a prosody diversifying module that is trained based on conditional DPP.

**Questions For The Authors:**

1. How are the predicted prosody features being added to the main TTS system? It says in the appendix but I could not find it.

**Reasons To Accept:**

1. The conditional DPP used for diverse prosody modelling is novel and
2. The derivations and explanations are plausible.
3. Experiments are adequate

**Reasons To Reject:**

1. The paper seems not complete: I can not find any appendices.
2. It is usually needed to have a demonstration webpage as TTS work needs subjective listening to demonstrate its superiority.

**Reproducibility:**

3: Could reproduce the results with some difficulty. The settings of parameters are underspecified or subjectively determined; the training/evaluation data are not widely available.

**Reviewer Confidence:**

4: Quite sure. I tried to check the important points carefully. It's unlikely, though conceivable, that I missed something that should affect my ratings.

---

> ### Author Rebuttal · Authors · 2023-08-26
>
> We thank the reviewer for the helpful comments for the paper.
>
> > The paper seems not complete: I can not find any appendices. How are the predicted prosody features being added to the main TTS system?
>
> The appendix is included in the Supplemenrary Materials. The usage of prosodic features is similar to FastSpeech2 [1]. Predicted duration, pitch an energy features are added to the text encoder output in the hidden dimensions.
>
> >It is usually needed to have a demonstration webpage as TTS work needs subjective listening to demonstrate its superiority.
>
> [The demo webpage](https://dpp-tts.github.io) is released. Audio samples from LJSpeech and the paragraph are provided.
>
> [1] FastSpeech 2: Fast and High-Quality End-to-End Text to Speech (https://arxiv.org/abs/2006.04558)

---

### Official Review · Reviewer_QLVm · 2023-08-04

**Soundness:** 3

**Excitement:**

3: Ambivalent: It has merits (e.g., it reports state-of-the-art results, the idea is nice), but there are key weaknesses (e.g., it describes incremental work), and it can significantly benefit from another round of revision. However, I won't object to accepting it if my co-reviewers champion it.

**Paper Topic And Main Contributions:**

The authors adopted DDP to improve prosody diversity in TTS, trying to improve perceptual prosodic diversity while maintaining naturalness and focusing on both the diversity of a single sample and among samples. Also, modified prosody predictor, prosodic boundary detector, and MIC objective function are proposed to ensure the effectiveness.

**Questions For The Authors:**

Quection A. In the subjective comparison of single sentences and paragraphs, how many audio samples were provided to the testers to score?

Quection B. I am confused with the function of Section 6 Related Work, also, since both the introduction and related work part mentioned works that using VAEs to improve prosody, probably apart from flow and diffusion based baselines, another baseline based on VAE may be useful to be included.

**Reasons To Accept:**

The authors attempt to adopt DDP to improve prosody diversity in TTS, which is not only applicable to single sample diversity but also to the diversity among samples. This is a novel approach.

Through subjective experiments, the authors demonstrate that previous TTS models that increase temperature do not result in perceptual prosodic diversity but rather reduce naturalness.

The model is well-designed, including predictors of duration and pitch, PDM module, prosodic boundary detection, and an adaptive MIC loss function. A series of ablation experiments validate the effectiveness of each module and the MIC objective function.

Experiments were implemented on various baselines, including Flowtron, VITS, DiffSpeech, which use Flow and Diffusion to enhance prosodic diversity, and Synta Speech. A series of ablation experiments have verified the effectiveness of the proposed modules and the MIC objective function.

**Reasons To Reject:**

The writing may need advision. For example, section 6 (Related Work) is highly similar to the Introduction's second paragraph.

There is room for improvement in the experimental details. For example,  additional quantitative experiments compared to baselines (Flowtron, DiffSpeech, or SyntaSpeech) could be considered. As mentioned in line 525, LJSpeech, as a single-speaker non-narrative dataset, may lack sufficient prosody patterns. Alternatively, the VCTK multi-accent dataset or the LibriTTS multi-person novel dataset may be considered.


**Reproducibility:**

4: Could mostly reproduce the results, but there may be some variation because of sample variance or minor variations in their interpretation of the protocol or method.

**Reviewer Confidence:**

3: Pretty sure, but there's a chance I missed something. Although I have a good feel for this area in general, I did not carefully check the paper's details, e.g., the math, experimental design, or novelty.

**Typos Grammar Style And Presentation Improvements:**

When the MIC objective function first appeared in the full text of Line 78, it may be better to be fully named.

---

> ### Author Rebuttal · Authors · 2023-08-26
>
> We thank the reviewer's thorough and constructive comments. Here are the responses.
>
> **Reply to reasons to Reject**
> >The writing may need advision. For example, section 6 (Related Work) is highly similar to the Introduction's second paragraph.
>
> Thank you for the feedback. We'll make sure to update the content so it doesn't overlap.
>
> >There is room for improvement in the experimental details.
>
> Here is an integrated quantitative evaluation including the baselines.
>
> | Model        | $\sigma_p$ | Pitch Determinant | Inference time  |
> |-------------|---------|---------------|-----------------|
> | Flowtron    | 0.0121  | $6.76 \cdot 10^{-14}$   | $5.3 \cdot 10^{-1}$       |
> | DiffSpeech  | 0.0116  | $8.51 \cdot 10^{-14}$   | $4.1 \cdot 10^{-2}$       |
> | SyntaSpeech | 0.0133  | $2.23 \cdot 10^{-14}$   | $3.9 \cdot 10^{-2}$       |
> | DPP-TTS-p   | 0.0145  | $1.88 \cdot 10^{-13}$   | $4.4 \cdot 10^{-2}$       |
>
> If the model does not have a separate duration predictor like VITS, it is challenging to evaluate the duration of phonemes, therefore only the pitch evaluation is included.
>
> One of the main reasons we chose the LJSpeech is to demonstrate our model's ability to generate diverse prosody patterns beyond the ground truth of non-narrative dataset. Even on a dataset which lacks diverse prosody pattern, if we can generate diverse prosodic features that do not exist in the dataset, it would serve as an effective demonstration of our model's capabilities.
> In addition, we also observe good performance in the paragraph evaluation which is not included in the LJSpeech. For the case of VCTK benchmark, and the audio samples will be included in the [Demo Webpage](https://dpp-tts.github.io/).
>
> **Reply to Questions**
> > In the subjective comparison of single sentences and paragraphs, how many audio samples were provided to the testers to score?
>
> In the case of the LJSpech dataset, 75 audio samples that are randomly sampled from the held-out dataset are evaluated and 10 testers are assigned to each sample. For the comparison of paragraphs,  5 long paragraphs which are not included in the LJSpeech are used for generating audio samples.  Likewise the LJSpeech dataset, 10 testers are assigned to each audio sample.
>
> >Since both the introduction and related work part mentioned works that using VAEs to improve prosody, probably apart from flow and diffusion based baselines, another baseline based on VAE may be useful to be included.
>
> Thank you for the suggestion. We experiment **VITS** because it is a sophisticated TTS model based on VAEs, consisting of a variational autoencoder backbone along with normalizing flow to enhance the prior.

---

### Official Review · Reviewer_gArD · 2023-08-10

**Soundness:** 3

**Excitement:**

4: Strong: This paper deepens the understanding of some phenomenon or lowers the barriers to an existing research direction.

**Paper Topic And Main Contributions:**

This paper proposes the Prosody Diversifying Module(PDM), which aims to generate a more diverse and smooth prosody pattern in text-to-speech system.

This paper analyzes the limitations of some of the current work on prosodic modeling in the field of speech synthesis, points out that sampling with high temperatures may degrades the naturalness of speech and these work treat each sample independently. This paper attempts to solve these problem by proposing a prosody diversifying module (PDM) which is based on conditional determinantal point processes (DPP). This module is utilized in tandem with a Fastspeech2-based TTS model.  In addition, the authors extracted key segments by using Prosodic Boundary Detector (PBD) to get a more accurate target. The paper mainly shows Mean Opinion Scores (MOS) comparing the proposed approach against four strong models for rhythmic and pitch.

**Reasons To Accept:**

1. To the best of my knowledge, the proposed method incorporating the PDM and DPP kernel represents a novel architecture compared to existing methods for the task of speech synthesis, despite the absence of a notable improvement.
2. The author addresses a pivotal issue in prosodic modeling within the domain of speech synthesis, and provides a comprehensive discussion on the challenges associated with prosodic modeling. For instance, the range of targets for conditional DPP sampling lacks clarity, and prosodic features typically exhibit variations in length.

**Reasons To Reject:**

The author's performance in the experiment is somewhat unsatisfactory.
1. Ablation studies are missing. The author highlights the fine-grained level of the prosody modeling method based on PBD and its contribution to diverse prosody in each generated speech sample. However, no ablation experiments focusing on the PBD component were conducted.
2. For unclear reasons, the ancillary materials only provide two types of samples from the baseline models, and not all of them are included.
3. In my perspective, the PDM appears to be a plug-and-play component in the TTS system. Similar recent work, such as CLAPSpeech [1], has explored prosody modeling through contrastive learning of multiple sampling. I am curious about the advantages of the author's complex structural design compared to such simple yet effective approaches in real-world scenarios.

[1] CLAPSpeech: Learning Prosody from Text Context with Contrastive Language-Audio Pre-training. 2023 ACL (https://arxiv.org/abs/2305.10763)

4. In Section 4, based on the experimental results presented in Table 1 for DPP-TTS-d/SyntaSpeech and DPP-TTS-p/SyntaSpeech, it is apparent that the MOS value of DPP-TTS is relatively low. However, the author has not provided an explanation for this observation. I am also curious about the author's interpretation of the MOS value of 4.09 ± 0.09 for DPP-TTS w/o PDM.
5. The level of persuasiveness in the conducted experiment is insufficient. The author should consider training and testing on larger and more diverse datasets. Additionally, relying solely on MOS-P and MOS-D as indicators of speech quality may not be comprehensive, and factors such as speech naturalness (despite the author's analysis of some unfavorable cases) should be taken into account. Furthermore, more experiments should be conducted to compare DPP-TTS and DPP-TTS w/o PDM, which would provide evidence of the wide range of functionalities offered by the PDM model.

**Reproducibility:**

4: Could mostly reproduce the results, but there may be some variation because of sample variance or minor variations in their interpretation of the protocol or method.

**Reviewer Confidence:**

3: Pretty sure, but there's a chance I missed something. Although I have a good feel for this area in general, I did not carefully check the paper's details, e.g., the math, experimental design, or novelty.

---

> ### Author Rebuttal · Authors · 2023-08-26
>
> We thank the reviewer for the constructive and detailed comments. Here are responses regarding the concerns.
>
> **Reply reasons to reject**
> >No ablation experiments focusing on the PBD component were conducted
>
> We conduct an experiment to demonstrate **the effectiveness of PBD over the fixed-length baseline**. While PBD *dynamically* adjust the length of the target considering prosodic boundary, the fixed-length baseline just extracts the target of fixed length.
>
> |      Model A/B      |   A  | Same |   B  | Model A MOS | Model B MOS |
> |:-------------------:|:----:|:----:|:----:|:-----------:|:-----------:|
> |  DPP-TTS-d/baseline | 58.2 | 24.5 | 17.3 |  4.02 $\pm$ 0.08  |  3.78 $\pm$ 0.06  |
> | DPP_TTS-p/baseline  | 61.8 | 23.4 | 14.8 |  3.98 $\pm$ 0.08  |  3.78 $\pm$ 0.06  |
> |                     |      |      |      |             |             |
>
> (baseline= DPP-TTS w/o PBD)\
> We can see from the table, both the perceptual diversity and MOS of the baseline decrease compared to the model using PBD. It indicates the advantage of PBD which dynamically adjusts the target length considering the prosodic boundary.
>
> >For unclear reasons, the ancillary materials only provide two types of samples from the baseline models, and not all of them are included.
>
> Some samples from Diffspeech and Syntaspeech are provided in the [Demo Webpage](https://dpp-tts.github.io/). We will update the supplementary materials so that all samples subject to the evaluation are included.
>
> >I am curious about the advantages of the author's complex structural design compared to such simple yet effective approaches in real-world scenarios.
>
> Although the goal of the work in [1] and our work are both prosody modeling, there exists a clear distinction between the goals.  The goal of the work in [1] is to generate samples similar to the ground truth samples as possible. In contrast, our research aims to explore the perceptual diversity of audio samples beyond mere imitation of the ground truth, once a baseline level of naturalness has been achieved.Our research aims to explicitly diversify prosody based on the context, following the generation of prosodic features by the baseline prosody predictor. This suggests that PBD module can be seamlessly integrated as a plug-and-play component. Besides, as the number of parameters of PDM is only 0.3M and the inference time is still reasonable due to the non-autoregressive prosody modeling, the module can be easily integrated into the other TTS model.
>
> >In section 4, it is apparent that the MOS value of DPP-TTS is relatively low compared to the Syntaspeech.
>
> We once again emphasize that our main goal is to achieve **better perceptual diversity**,  maintaining the naturalness of audio samples. We consider that the audio samples have achieved perceptual diversity if they show good performance in the side-by-side evaluation and the MOS does not degrade too much. When audio samples show good performance in the side-by-side evaluation and the MOS does not decrease too much, we consider that the audio samples have achieved perceptual diversity. The MOS value of Table 1 is served as supplementary metric since big MOS differences between samples may bias listeners to one sample without considering the diversity. Actually, if we increase the quality weight, it is possible to achieve a better MOS sample over the Syntasepech:
> | Model A/B             | Model A MOS | Model B MOS |   |   |
> |-----------------------|-------------|-------------|---|---|
> | Dpp-TTS-d/Syntapeech  | 4.07 $\pm$ 0.07   | 4.04 $\pm$ 0.09   |   |   |
> | Dpp-TTS-p/Syntaspeech | 4.09 $\pm$ 0.07   | 4.04 $\pm$ 0.09   |   |   |
> |                       |             |             |   |   |
>
> If we are merely aiming for naturalness, we can achieve better MOS at the cost of losing some diversity.
>
> >The author should consider training and testing on larger and more diverse datasets.
>
> We are conducting an additional experiment on the VCTK benchmark, and we will merge the results in the future manuscript and the [Demo webpage](https://dpp-tts.github.io). However, note that we also presented paragraph evaluation outside of the LJSpeech to demonstrate the model's generalization ability. Performing well on longer textual data sets,  which is different from LJSpeech, suggests that our model is not limited to a specific datasets.
>
> >Additionally, relying solely on MOS-P and MOS-D as indicators of speech quality may not be comprehensive, and factors such as speech naturalness should be taken into account.
>
> For the evaluation of speech naturalness, the MOS-P metric has been used in many literatures [2,3,4]. Since the main focus of our research is the prosody aspect of speech, we stick to the prosody aspect of MOS rather than the original MOS.
>
> > Furthermore, more experiments should be conducted to compare DPP-TTS and DPP-TTS w/o PDM
>
> We present an experiment result regarding the **quality weight $w$** of PDM.
>
> | Model A/B              | A    | Same | B    | Model A MOS | Model B MOS |
> |------------------------|------|------|------|-------------|-------------|
> | DPP-TTS/DPP-TTS-w=1.0  | 24.5 | 17.7 | 57.8 | 4.02 $\pm$ 0.08   | 3.95 $\pm$ 0.09   |
> | DPP-TTS/DPP-TTS-w=5.0  | 44.5 | 24.2 | 31.3 | 4.02 $\pm$ 0.08   | 4.07 $\pm$ 0.06   |
> | DPP-TTS/DPP-TTS-w=10.0 | 57.6 | 19.5 | 22.9 | 4.02 $\pm$ 0.08   | 4.11 $\pm$ 0.06   |
>
> DPP-TTS with quality weight $w=2.0$ is used for Model A. As the table demonstrates, we can trade off between diversity and naturalness without having a severe impact on naturalness.  These experimental observations sufficiently demonstrate the effectiveness of our methodology.
>
> [1] CLAPSpeech: Learning Prosody from Text Context with Contrastive Language-Audio Pre-training. 2023 ACL (https://arxiv.org/abs/2305.10763) \
> [2] PortaSpeech: Portable and High-Quality Generative Text-to-Speech. 2021 Neurips (https://arxiv.org/abs/2109.15166)  \
> [3] SyntaSpeech:  Syntax-Aware Generative Adversarial Text-to-Speech. 2022 IJCAI (https://arxiv.org/abs/2204.11792) \
> [4] Prosody-TTS: Improving Prosody with Masked Autoencoder and
> Conditional Diffusion Model For Expressive Text-to-Speech. 2023 ACL (https://aclanthology.org/2023.findings-acl.508.pdf)

---

### Meta-Review · Area_Chair_w47N · 2023-09-19

**Recommendation:** 5

**Metareview:**

This paper identifies the limitation of prior work on generating diversity prosody (pitch and duration) patterns, and presents a plug-and-play prosody diversifying module (PDM) to effectively tackle the issue. All reviewers agree that the authors address an important issue in speech synthesis, present a novel DPP-based solution that is well-designed and thoroughly evaluated when combined with various TTS backbones.

The authors have presented many additional studies and samples during the rebuttal period, which sufficiently addressed reviewers concern on ablation studies. These results would be valuable to the readers and I suggest authors to incorporate them into the paper.

---

### Decision · Program_Chairs · 2023-10-07

**Decision:**

Accept-Main

**Comment:**

This paper identifies the limitation of prior work on generating diversity prosody (pitch and duration) patterns, and presents a plug-and-play prosody diversifying module (PDM) to effectively tackle the issue. All reviewers agree that the authors address an important issue in speech synthesis, present a novel DPP-based solution that is well-designed and thoroughly evaluated when combined with various TTS backbones.

The authors have presented many additional studies and samples during the rebuttal period, which sufficiently addressed reviewers concern on ablation studies. These results would be valuable to the readers and I suggest authors to incorporate them into the paper.